# Ménière's Disease: Insights from an Italian Nationwide Survey

Fulvio Mammarella [1], Antonella Loperfido [1], Elizabeth G. Keeling [2], Gianluca Bellocchi [1] and Luca Marsili [3,*]

[1] Otolaryngology Unit, San Camillo Forlanini Hospital, 00152 Rome, Italy
[2] School of Life Sciences, Arizona State University, Tempe, AZ 85281, USA
[3] Gardner Family Center for Parkinson's Disease and Movement Disorders, Department of Neurology, University of Cincinnati, Cincinnati, OH 45219, USA
* Correspondence: luca.marsili@uc.edu; Tel./Fax: +1-(513)-558-7643

**Abstract:** The aim of the present study was to obtain data from a large community sample of patients with Ménière's disease (MD) in Italy through a web-based nationwide survey. Demographic, clinical, and epidemiological features of MD among members of the Italian Association of Ménière's Disease (AMMI) were collected through a web-based survey. The questionnaire was posted on the AMMI website between 01/SEP/2021 and 31/OCT/2021. A total of 520 patients (374 F, 146 M) with MD were included. The age at interview (average ± standard deviation, SD) was 51.4 ± 10.9 years, with a disease duration of 9.9 ± 9.8 years. Eighty percent of cases were unilateral. No patients reported neurocognitive disorders or Parkinson's disease. A positive family history of MD was reported in 13% of participants, while a history of allergic diseases was reported in 33%. Comorbid thyroid disorders were present in 25% of participants, and 28% used betahistine as the main treatment. To our knowledge, this is the first study that has investigated the epidemiology and current patterns of care of MD in Italy, using an anonymous survey directly sent to patients, thus implying their active participation. We hope that future studies will support the utilization of web-based surveys to address the unmet needs in the management of patients with MD.

**Keywords:** Ménière's disease; survey; otologic disorders

## 1. Introduction

Ménière's disease (MD) is a pathological condition of the inner ear characterized by the clinical association of dizziness, hearing loss, fullness, and tinnitus [1,2]. The pathophysiology of MD is still unclear. Some studies have suggested that MD is associated with the accumulation of endolymph in the cochlea and vestibular organs, although endolymphatic hydrops (EH) per se does not explain all clinical features, including the progression of hearing loss or the frequency of attacks of vertigo [1–4]. It is still unknown whether the cochlea is truly the beginning shock organ; in fact, vestibular MD may present with vertigo and without hearing loss [5].

EH consists of an excessive accumulation of endolymphatic fluid within the scala media, which causes intracochlear distension of the Reissner's membrane and of all endolymphatic compartments [6]. MD typically causes both cochlear and vestibular symptoms and affects only one side (bilateral MD is a less common condition) [7]. MD's etiology is not yet known, and anatomical investigations in this regard have only been conducted on a few post-mortem cases. Recent studies report the histopathological features of this condition, describing vestibular neuroepithelium damage with hair cell loss, basement membrane thickening, and perivascular microvascular damage [8]. Other authors have suggested a pathological correlation with the decreased expression of a mitochondrial protein import receptor in the human cochlea called Tom20, a translocase of the outer mitochondrial membrane [3]. This hypothesis suggests the possibility of a combination between neurodegenerative and ear affections, possibly configuring a benign disease subtype, as recently postulated in Parkinson's disease [9].

Epidemiological data demonstrate a higher prevalence in white females (F: M ratio = 1.25), with a mean age of onset of 46.5 years [10]. MD is almost always unilateral [1,11]. Familial bilateral conditions are characterized by an autosomal dominant inheritance genetic pattern with incomplete penetrance and evidence of anticipation, or a more severe phenotype, in offspring [12,13]. However, familial MD can also be unilateral [14]. There are other diseases with different etiologies that can mimic MD's symptoms, such as Cogan syndrome and vestibular migraine, requiring a careful clinical and instrumental evaluation (e.g., video-head impulse test, caloric test, skull-vibration test) for the differential diagnosis [15,16].

Epidemiological investigations on the MD in the last two decades have been scanty, although in the last few years there have been some reports on the topic [17–19]. The aim of the present study was to obtain data from a large community sample of patients with MD in Italy through a web-based nationwide survey, to better elucidate the epidemiology and the current patterns of care for MD, and finally to identify possible areas needing increased awareness among both patients and physicians.

## 2. Materials and Methods

We conducted a nationwide survey on the epidemiological features of MD among members of the *Associazione Malati di Ménière Insieme* (AMMI) who were willing to take part in the survey. The AMMI is a non-profit organization composed of patients with MD that provides education, awareness, support, and resources for persons with MD. The participants (all diagnosed with MD by a physician—this is a requirement to be a member of the AMMI) were freely invited to participate in the survey, which was made easily accessible through a specific link on the website's homepage and advertised by the AMMI. The compilation of the questionnaire was anonymous (e.g., all data were de-identified), and the only personal data included the sex and age of the participant (month and year of birth). Participants were given the option to complete the questionnaire with the help of a caregiver, if needed. The questionnaire was posted on the AMMI website for sixty days, from 1 September 2021 to 31 October 2021. Inclusion criteria in the survey entailed patients being members of the AMMI and correctly filling out all the items proposed (a maximum of two non-replied multiple-choice questions was allowed), whereas exclusion criteria were the inaccurate and incomplete compilation of the proposed items. The questionnaire was created and proposed by the authors of the present paper in the Italian language, and it was subjected to prior evaluation and authorization by the AMMI Association's Chair and a local related Institutional Review Board. Figure 1 shows the questionnaire translated into English, which comprises a total of eighteen items: five single-choice and thirteen multiple-choice questions, including age at diagnosis, involvement of one or both ears, injured side, need for hearing aids, number of dizziness episodes per month, need for walking assistance devices, history of migraine or headache, presence of other first-degree family members diagnosed with MD, concomitant neurological diseases, other comorbidities or allergies, and drugs used by the patient. For the compilation, the association's webmaster created an appropriate Italian format with the aim of developing a quick and easy questionnaire that required a maximum of five minutes for compilation. The association's webmaster made the data analysis using Excel tables and created dedicated graphics for each question. The data were subsequently grouped by the investigators into descriptive (e.g., age, sex, year of diagnosis, etc.) and comparative (e.g., one or two affected sides) statistics.

Sex M/F
Date of birth

Questions:

1. When was Ménière disease diagnosed (year)?
2. Your Ménière disease is unilateral or bilateral? Specify
3. If unilateral, which side? Specify
4. Do you use acoustic prothesis? Y/N
5. Do you have five or more episodes of dizziness/vertigo per month? Y/N
6. Do you take any drugs for Ménière disease? Y/N
7. Do you suffer from headache/migraine? Y/N
    a. If yes, please specify:
8. Do you have any family member diagnosed with Ménière disease? Y/N
    a. If yes, please specify:
9. Do you suffer from any allergies? Y/N
    a. If yes, please specify:
10. Do you suffer from any other disorders of the ear, nose and throat? Y/N
    a. If yes, please specify:
11. Have you been diagnosed with Parkinson's disease? Y/N
    a. If yes, please specify (year of diagnosis, main symptoms):
12. If yes, do you have a neurologist provider? Y/N
    a. If yes, please specify which drugs are you taking for Parkinson's disease
13. Do you suffer from cognitive impairment or dementia? Y/N
    a. If yes, please specify (Alzheimer's, other cognitive disorders, another dementia
       type, mild cognitive impairment):
14. If yes, please specify which drugs are you taking for your cognitive impairment /
    dementia:
15. Do you use a walker or any device to help you walking (e.g., cane, etc.)?
    a. If yes, please specify:
16. Have you had accidental falls in the last year? Y/N
    a. If yes, please specify the total number (or estimated total number)
17. Do you suffer from any other medical conditions? Y/N
    a. If yes, please specify:
18. Do you take any other drugs?
    a. If yes, please specify:

**Figure 1.** Questionnaire for Ménière's disease.

## 3. Results

A total of 520 patients (374 females, 72% of the sample) with MD responded to our survey and were included in the study. Across the cohort, the age at interview (average ± standard deviation—SD) was 51.4 ± 10.9 years (range 26–82 years), with a disease duration (average ± SD) of 9.9 ± 9.8 years (range 1–65 years). Eighty percent of cases were unilateral (415/520), whereas 20% were bilateral (105/520) [11]. When unilateral, MD involved the left side in 52% (215/520) of cases and the right side in 48% (201/520) of cases (Figure 2A–C).

Sixteen percent (84/520) of participants declared the use of acoustic protheses. Additionally, 16% (82/520) of participants reported accidental falls during the previous year. Forty-nine percent (247/520) of patients reported comorbid headache (Figure 2D–F). No patients reported neurodegenerative disorders such as dementia or cognitive impairment, nor Parkinson's disease or parkinsonian disorders; only two patients reported cases of parkinsonism in their families (the father of a patient in one case and someone in the mother's family in the second case). Accordingly, no patients reported using walking aids, except for three who used canes and one who used a walker. Positive family history of MD was reported in 13% (67/520) of participants, whereas the remaining 87% (453/520) did not report any affected family members. A history of allergic diseases was reported in 33% of cases (173/520), whereas 67% (347/520) of cases did not report allergies. Nine percent (48/520) of patients reported other otolaryngological disorders (e.g., sinusitis, rhinitis, hypertrophy of the turbinates, nasal polyposis, laryngitis/pharyngitis, otitis, tonsil disorders, otosclerosis), whereas 91% (472/520) did not (Figure 3A–D). A total of 41% (214/520) of patients reported other disorders in association with MD, and 55% (289/520) did not (17 participants did not provide answers to this specific question). Out of the 214 patients with comorbid disorders, 95% (204) provided accurate data. The reported diseases were thyroid

disorders in 25% (51/204), osteoarthritis in 13% (26/204), and cardiovascular diseases in 13% (27/204) of responders. Other comorbidities are reported in Figure 4A,B. Regarding drugs for MD, 474 patients provided accurate responses to the questionnaire. Specifically, 28% (132/474) of participants used betahistine, followed by 26% (121/474) treated with diuretics (any type), 11% (54/474) with calcium-antagonists (e.g., flunarizine, cinnarizine), and 8% (37/474) and 6% treated with SSRIs or benzodiazepines, respectively. Drugs taken for other diseases included levothyroxine in 31% (52/168), antihypertensive drugs in 20% (34/168), analgesics in 12% (20/168), and statins in 10% (17/168) of participants (Figure 5).

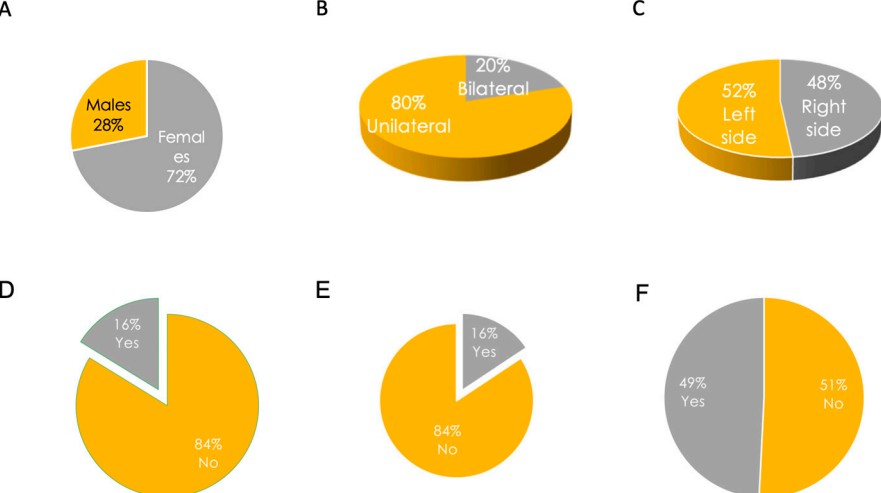

**Figure 2.** Characteristics of participants. (**A**) Sex of participants; (**B**) Laterality of Ménière's disease; (**C**) Proportion of sides (**left** versus **right**) when unilateral; (**D**) Percentage of participants using acoustic protheses; (**E**) Percentage of participants with accidental falls in the previous year; (**F**) Percentage of participants with comorbid headache.

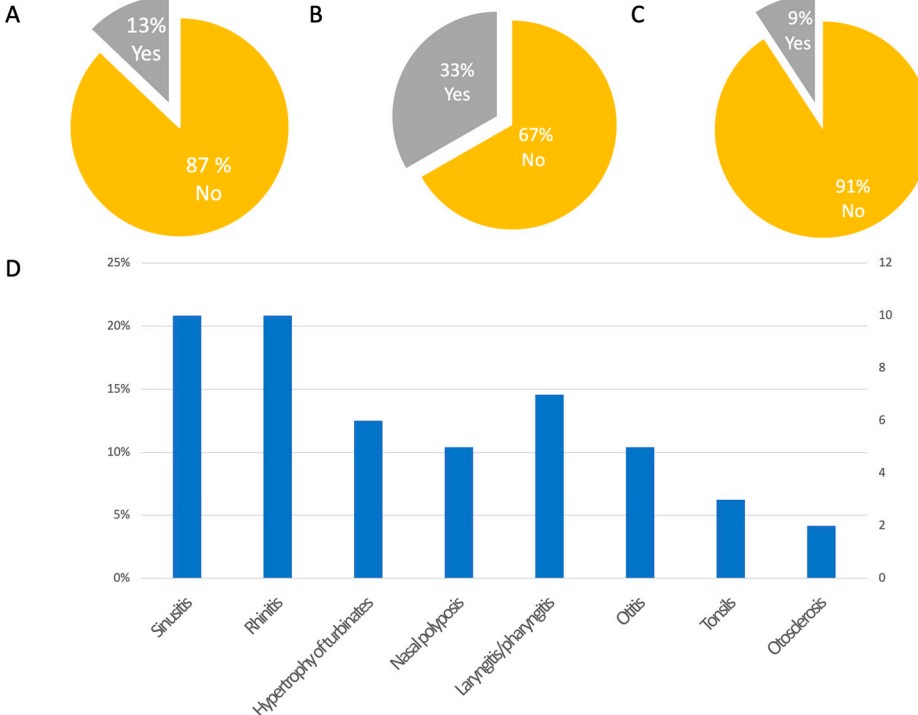

**Figure 3.** Other epidemiological features. (**A**) Percentage of participants with family history of Ménière's disease; (**B**) Percentage of participants with comorbid allergies; (**C**) Percentage of participants with other otolaryngological disorders; (**D**) Type of associated otolaryngological disorders.

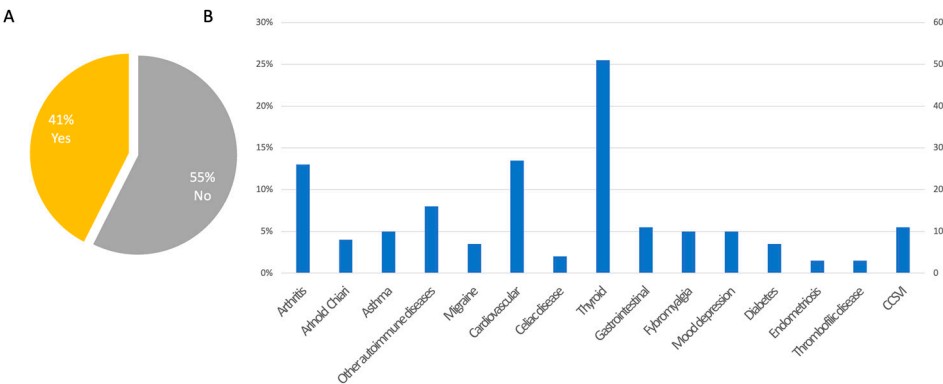

**Figure 4.** Comorbidities. (**A**) Percentage of any other comorbidly associated disorders; (**B**) Type of comorbidly associated disorders.

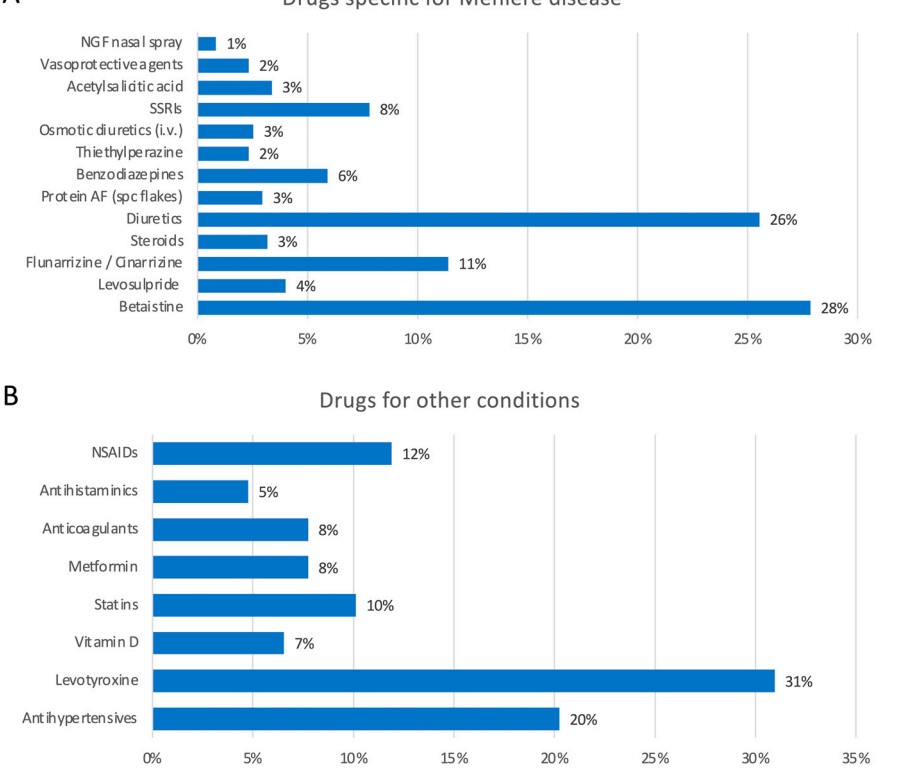

**Figure 5.** Drugs. (**A**) Drugs used to treat Ménière's disease and their percentage; (**B**) Drugs used to treat any other conditions.

## 4. Discussion

To our knowledge, this is the first study in the literature that has investigated the epidemiology and the current patterns of care of MD in Italy. By using an anonymous survey directly sent to patients, we have confidence in their active participation and honest responses, also due to the significant motivation to improve clinical care for MD patients who are currently members of an association such as the AMMI. Surveys represent a simple, quick-to-execute, and targeted research methodology on a selected sample population that can be performed at any time and in any place using self-completed questionnaires. They allow a transversal collection of data, overcoming the limits and most common errors of mono- and multicentric retrospective studies [20]. The survey as a data collection method is widely used in various scientific fields but is still rare in the field of otolaryngology. A previous multicentric survey on familial clustering and genetic heterogeneity in MD,

carried out by Requena and colleagues, supported the idea that MD is characterized by a strong familial aggregation and that sporadic and familial MD are clinically identical [21].

Our data analysis shows a female prevalence in MD, in accordance with the literature [22]. The data on disease onset had a negatively skewed distribution, with the vast majority of patients diagnosed between 0 and 8 years before the survey (83% of patients were diagnosed between 3 and 4 years prior to the survey, whereas only 7 patients were diagnosed between 37 and 42 years prior to the survey). More in detail, the age at onset of MD in our sample was 41.7 ± 5.8 years (range 17–69 years), and the average age of the patients interviewed was similar to that of the most recent studies [23–26] (Supplementary Figures S1 and S2). As widely reported in the literature, most cases were unilateral [27]. It could be possible that the percentage of bilateral disease is low because of the short duration of the follow-up period, approximately 7.6 years [28]. In our survey, a positive family history of MD was reported by 13% of participants, a percentage that is slightly higher than the average described in other studies [22,25,29,30]. The use of acoustic protheses is still infrequent, with only 16% of the sample using this aid. The difficulties in the application of hearing aids in MD include the presence of fluctuating and commonly asymmetrical hearing loss, a rising audiometric configuration, a reduced dynamic range, and reduced word-recognition scores [30].

Therapeutic options for unilateral hearing loss include not only conventional hearing aids but also contralateral routing of sound (CROS), osteointegrated hearing aids, and, in more severe cases, cochlear implants [31]. From the results collected, thyroid diseases were the most common comorbidities reported (all types of thyroiditis, 25% of patients reporting comorbidities). A recent study shows that the association of MD with thyroid disorders may be due to autoimmunity and dysregulated thyroid hormone levels in these patients [32]. Other reported comorbidities were cardiovascular diseases (hypertension, followed by arrhythmias, 13%), arthritis (13%—mainly rheumatoid arthritis), and other autoimmune diseases (psoriasis, multiple sclerosis, and systemic lupus erythematosus—S.L.E., 8%; celiac disease, 2%). Thirty-three percent of our participants reported the coexistence of allergic conditions, especially airborne allergies, consistent with the literature [33]. The association between migraine and MD, which is well known, is confirmed in about half of the population interviewed, in accordance with the international literature [33].

Overall, our findings are in line with the recent studies published by the Spanish group of Frejo and colleagues, which reported five main clusters of bilateral MD symptoms: metachronic hearing loss, synchronic hearing loss, familial MD, MD with migraine, and MD with autoimmune disease [26], and then compared these results in patients with unilateral MD [11]. The age of onset of MD in the studies by Frejo et al. [11,26] ranged from 39 ± 12.9 (bilateral)–45.3 ± 13.6 (unilateral) years in familial MD and 44.8 ± 13.1 (bilateral)–45.6 ± 12.5 (unilateral) in sporadic MD, in line with our results (Supplementary Figure S3). The comorbidity with headache ranged from 44% (bilateral)–40.2% (unilateral) in familial MD and 36.1% (bilateral)–32.1% (unilateral) in sporadic MD, similar to what we reported. Autoimmune disorders were highly reported in both studies by Frejo and colleagues, ranging from 11% to 19.4% in unilateral sporadic and familial MD, respectively; rheumatoid arthritis, followed by psoriasis, thyroid disorders, and SLE, were the most frequently reported autoimmune diseases in bilateral MD [11,26], again in line with our results. Among other comorbidities, the most frequently reported conditions were cardiovascular diseases, and arterial hypertension was the most represented condition, ranging from 26.5% (familial) to 39.7% (sporadic) in bilateral MD [11,26], slightly elevated when compared to our sample, probably due to the younger age at examination of our patients.

On the other hand, no patients reported neurodegenerative disorders such as mild cognitive impairment, any types of dementia, or parkinsonism. This result could be underestimated due to the bias related to the survey method used and the relatively young average age of participants (e.g., 51 years old), which made it difficult to detect a disease most prevalent in the geriatric population [34]. However, this finding could also be due to the benign neurodegenerative disease course associated with MD, which could have somewhat masked the presence of the main neurological symptoms [9,35].

Of interest, 11% of MD patients were also taking flunarizine or cinnarizine, which are associated with tardive parkinsonism [36]. More studies are required to disentangle the interesting association between MD and parkinsonism. Sixteen percent of participants reported accidental falls during the previous year, with an average of up to three falls per year, in line with the conclusions of previous studies [37]. In fact, many authors report that dizziness is a recognized risk factor for falls, with consequent impact on the patients' quality of life [38].

Regarding treatment, we found heterogeneous results in terms of molecules, dosages, and posology. Treatment commonly included betahistine, as it was recently described in a systematic review [39]. Other reported treatments were diuretics, calcium-antagonists such as flunarizine and cinnarizine, SSRIs, and benzodiazepines, in line with the international literature [40].

Our study has some limitations, including the cross-sectional design that does not allow follow-up examinations, the intrinsic bias due to the self-administration of the survey, and the lack of the physician's confirmation of the diagnosis and provided data. We did not find a specific association between MD and neurodegenerative disorders, probably due to the self-administration of the survey and the relatively young age of the interviewed population, despite giving the option to complete the questionnaire with the help of a caregiver. Further longitudinal studies will need to be conducted on broader populations of patients to better investigate this possible association. Finally, due to the way we collected and analyzed the data, further comparisons with previous interesting studies on the phenotypic characteristics of MD [11,26], and mainly cluster analyses, were not possible. For the same reason, no further comparisons of the incidence of allergies or family history in different geographical areas of Italy were possible.

Surveys are powerful tools to conduct research in the current era; they are inexpensive, rapid, easily accessible to the patient population, and allow the active participation of the examined subjects. With the increasing expansion of technology and access to internet-based services, web-based surveys are creating a paradigm shift in the field of research. They allow a more active role for patients in gathering scientific data, thus making scientific research universally available and democratic. In the last few years, more attention has been given to the active participation of patients in research studies and clinical trials, and clinical scales are now considering patient-reported outcomes as fundamental parts of their clinimetric evaluations [41].

With the present study, we have provided updated information on epidemiology and patterns of care for MD from a sample of five hundred and twenty patients in Italy, collected through a web-based survey. In sum, we hope that future studies supported by national and supranational governmental agencies will encourage the utilization of further web-based surveys to address the unmet needs in the management of patients with MD.

**Supplementary Materials:** The following supporting information can be downloaded at: https://www.mdpi.com/article/10.3390/audiolres13020016/s1, Figure S1: Year of diagnosis; Figure S2: Age distribution; Figure S3: Age at onset.

**Author Contributions:** Conceptualization, F.M. and L.M.; methodology, F.M., A.L. and L.M.; data curation, F.M., A.L., E.G.K. and L.M.; writing—original draft preparation, F.M. and L.M.; writing—review and editing, F.M., A.L., E.G.K., G.B. and L.M. All authors have read and agreed to the published version of the manuscript.

**Funding:** This research received no external funding.

**Institutional Review Board Statement:** All study procedures were performed in accordance with the ethical standards of the 1964 Declaration of Helsinki and its later amendments. The authors confirm that they have read the journal's position on issues involved in ethical publication and affirm that this work is consistent with those guidelines. Ethical approval was waived by the local Ethics Committee (Rome, Italy) in view of the retrospective nature of the study (not applicable to a single center and consisting of a survey addressed to collect de-identified data of volunteering subjects who

are members of a specific patients' association). However, the AMMI's internal ethics committee approved the study.

**Informed Consent Statement:** Not applicable.

**Data Availability Statement:** L. Marsili and F. Mammarella had full access to all the data in the study and take responsibility for the integrity of the data, the accuracy of the data analysis, and the conduct of the research. They have the right to publish any and all data, separate and apart from the guidance of any sponsor.

**Acknowledgments:** We thank all patients who participated in the survey and the Italian Association of Ménière's Disease (AMMI; Associazione Malati di Ménière Insieme) for the help and support.

**Conflicts of Interest:** The authors declare that they have no conflict of interest related to the research covered in this article. Dr. Luca Marsili has received honoraria from the International Association of Parkinsonism and Related Disorders (IAPRD) Society for social media and web support. The other authors do not report any conflicts of interest.

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
