# Peer review of "Ménière’s Disease: Insights from an Italian Nationwide Survey"

_audiolres, doi:10.3390/audiolres13020016_

Round 1

Reviewer 1 Report

The formula of survey may be important to probe the situation, although a potential limitation is the lack of control. Admittedly I have seen some patients diagnosed elsewhere as MD which didn’t present all characters of MD [may be included in a more clear way in the potential limitations].

So that the way questionnaire is built is the centre of the work. There is an emphasis on Parkinson and Dementia that as far as I know are not correlated with MD, while for example other conditions are poorly mentioned (hypertension, vascular disorders among others, prtially autoimmune disorders rather than Hashimoto). I think that on this point few can be done. I suggest anyway to correct other points. I've another cuiosity: data of the onset of vertigo presents a normal distribution?

Among references I would include the Barany Society Criteria 

 Moreover, few words on characterization of subgroup of MD (the  following work)

-       Extended phenotype and clinical subgroups in unilateral Meniere disease: A cross-sectional study with cluster analysis Frejo L, Martin-Sanz E, Teggi R, Trinidad G, Soto-Varela A, Santos-Perez S, Manrique R, Perez N, Aran I, Almeida-Branco MS, Batuecas-Caletrio A, Fraile J, Espinosa-Sanchez JM, Perez-Guillen V, Perez-Garrigues H, Oliva-Dominguez M, Aleman O, Benitez J, Perez P, Lopez-Escamez JA; Meniere's disease Consortium (MeDiC). Clin Otolaryngol. 2017 Dec;42(6):1172-1180. doi: 10.1111/coa.12844. Epub 2017 Feb 26.PMID: 28166395

-        In this recent work you will find overlapping data about comorbidities (Hashimoto included). I suggest to read and possiblu include among references

-       Evaluation of a large cohort of adult patients with Ménière's disease: bedside and clinical history. Teggi R, Battista RA, Di Berardino F, Familiari M, Cangiano I, Gatti O, Bussi M. Acta Otorhinolaryngol Ital. 2020 Dec;40(6):444-449. doi: 10.14639/0392-100X-N0776.

Did you receive an Ethics Committee approval? If not, please explicit why not necessary

Line 28: include fullness

Line 29: I would rather write in the cochlea. Citations 1-3, I suggest to include Pathophysiology of Meniere's syndrome: are symptoms caused by endolymphatic hydrops? Merchant SN, Adams JC, Nadol JB Jr.

Line 32 and following: nothing is sure in pathophysiology of MD, including if cochlea is beginning shocK organ. It is reported that some MD begin with vestibular symptoms alone.

Line 47: familial cases may be also monolateral

Line 51: perhaps you can find other references. There’s often a dissociation between video HIT and calorics or with skull vibration test.

Line 61: did you check correctness of diagnosis?

Line 81: which parental degree was considered?

Author Response

Comment: The formula of survey may be important to probe the situation, although a potential limitation is the lack of control. Admittedly I have seen some patients diagnosed elsewhere as MD which didn’t present all characters of MD [may be included in a more clear way in the potential limitations].

So that the way questionnaire is built is the center of the work. There is an emphasis on Parkinson and Dementia that as far as I know are not correlated with MD, while for example other conditions are poorly mentioned (hypertension, vascular disorders among others, partially autoimmune disorders rather than Hashimoto). I think that on this point few can be done. I suggest anyway to correct other points. I've another curiosity: data of the onset of vertigo presents a normal distribution?

Answer: We thank the Reviewer for the general positive comment. We have now better stated in the “Discussion” section, that among the 41% of patients which reported comorbid diseases, 25% had comorbid thyroid disease (all types of thyroiditis), followed by 13% reporting cardiovascular diseases (which included hypertension, followed by arrhythmias), 13% reporting arthritis, and finally by 10% reporting other autoimmune diseases (8%, mainly represented by psoriasis, followed by multiple sclerosis and S.L.E.) and celiac disease (2%) (Lines: 178-184).

Also, data of disease onset had a negatively skewed distribution with the vast majority of patients diagnosed between 0 and 8 years before the survey. More specifically, 83% of patients were diagnosed between 3 and 4 years prior to the survey, whereas only 7 patients were diagnosed between 37 and 42 years prior (Lines: 160-164; Supplementary Tables 1 and 2).

Comment: Among references I would include the Barany Society Criteria 

            Answer: We thank the Reviewer for the insightful comment. We have now updated the diagnostic criteria, accordingly (Line: 29)

Comment:  Moreover, few words on characterization of subgroup of MD (the following work)

 Extended phenotype and clinical subgroups in unilateral Meniere disease: A cross-sectional study with cluster analysis Frejo L, Martin-Sanz E, Teggi R, Trinidad G, Soto-Varela A, Santos-Perez S, Manrique R, Perez N, Aran I, Almeida-Branco MS, B            atuecas-Caletrio A, Fraile J, Espinosa-Sanchez JM, Perez-Guillen V, Perez-Garrigues H, Oliva-Dominguez M, Aleman O, Benitez J, Perez P, Lopez-Escamez JA; Meniere's disease Consortium (MeDiC). Clin Otolaryngol. 2017 Dec;42(6):1172-1180. doi: 10.1111/coa.12844. Epub 2017 Feb 26.PMID: 28166395.

In this recent work you will find overlapping data about comorbidities (Hashimoto included). I suggest to read and possibly include among references

                  Answer: We thank the Reviewer for suggesting this interesting article that we did not consider before. We have now added it in the reference list, cited it into the text and compared our results with those reported in that study (Lines: 50, 101; and 188-204).

Comment: Evaluation of a large cohort of adult patients with Ménière's disease: bedside and clinical history. Teggi R, Battista RA, Di Berardino F, Familiari M, Cangiano I, Gatti O, Bussi M. Acta Otorhinolaryngol Ital. 2020 Dec;40(6):444-449. doi: 10.14639/0392-100X-N0776.

      Answer: We have added the suggested reference in the Discussion section (Lines: 164, 169).

Comment: Did you receive an Ethics Committee approval? If not, please explicit why not necessary.

            Answer: We thank the Reviewer for making this important point. Accordingly, we have now specified in the dedicated section “Institutional Review Board Statement” that: “Ethical approval was waived by the local Ethics Committee (Rome, Italy) in view of the retrospective nature of the study (not applicable to a single center and consisting of a survey addressed to collect de-identified data of volunteers, member of a specific patients’ association). However, the AMMI internal ethics committee approved the study.” (Lines:258-262).

Comment: Line 28: include fullness

            Answer: We have added the term, as required (Line: 29).

Comment: Line 29: I would rather write in the cochlea. Citations 1-3, I suggest to include Pathophysiology of Meniere's syndrome: are symptoms caused by endolymphatic hydrops? Merchant SN, Adams JC, Nadol JB Jr.

            Answer: We have modified the sentence, as required (Lines: 29-31). The reference has been added (Line: 33).

Comment: Line 32 and following: nothing is sure in pathophysiology of MD, including if cochlea is beginning shocK organ. It is reported that some MD begin with vestibular symptoms alone.

            Answer: We agree with the Reviewer’s observation, and we have now rephrased the first paragraph of the introduction, accordingly (Lines: 28-35).

Comment: Line 47: familial cases may be also monoliteral

            Answer: We have added this important detail, as highlighted by the Reviewer (Line: 53).

Comment: Line 51: perhaps you can find other references. There’s often a dissociation between video HIT and caloric or with skull vibration test.

            Answer: We agree with the Reviewer’s comment and we have now modified the manuscript accordingly and added a new reference (Ref#16) (Lines: 55-57)

Comment: Line 61: did you check correctness of diagnosis?

            Answer: We thank the Reviewer for this important comment. We agree that the diagnostic confirmation is an intrinsic limitation of volunteers-based surveys and that we could not check or confirm it, due to the necessity of respecting the privacy rules and the anonymity of participants. Accordingly, we have added this aspect to the Limitations section of the discussion (Line: 225).

Comment: Line 81: which parental degree was considered?

            We have now better specified that the parental degree that we considered was the first degree. (Line: 86).

Reviewer 2 Report

This is a cross sectional survey performed with the Italian association of Meniere disease patients including 520 individuals. It is the first study on epidemiology self-reported data and it should be published.

Results

I will recommend the authors to compare their results with the studies published by Frejo et al. in 2016 and 2017 in Spain that included both around 1500 patients to obtain the distribution of the main comorbidities such as migraine or autoimmune disorders. There are 5 major clinical subgroups in Meniere disease and it should be interesting to obtain the relative frequency of each of them.

The authors should considering to add a chart bar with the age distribution of the patients and if possible the age of onset  of MD (revise Frejo el al. and compare the data).

Minor questions

The authors report familial history of 13% and allergic diseases  in 33%. Could the authors compare the differences north/south within Italy, since familial distribution could be different in north and south and also allergies since the respiratory allergies are related to humidity and temperatures on dry and wet seasons (Milan vs Rome).

The font size should be improved in Figure 3D and 4B, it is too tiny.

Suggested references

Frejo L, Martin-Sanz E, Teggi R, Trinidad G, Soto-Varela A, Santos-Perez S, Manrique R, Perez N, Aran I, Almeida-Branco MS, Batuecas-Caletrio A, Fraile J, Espinosa-Sanchez JM, Perez-Guillen V, Perez-Garrigues H, Oliva-Dominguez M, Aleman O, Benitez J, Perez P, Lopez-Escamez JA; Meniere's disease Consortium (MeDiC). Extended phenotype and clinical subgroups in unilateral Meniere disease: A cross-sectional study with cluster analysis. Clin Otolaryngol. 2017 Dec;42(6):1172-1180. doi: 10.1111/coa.12844. Epub 2017 Feb 26. PMID: 28166395.

Frejo L, Soto-Varela A, Santos-Perez S, Aran I, Batuecas-Caletrio A, Perez-Guillen V, Perez-Garrigues H, Fraile J, Martin-Sanz E, Tapia MC, Trinidad G, García-Arumi AM, González-Aguado R, Espinosa-Sanchez JM, Marques P, Perez P, Benitez J, Lopez-Escamez JA. Clinical Subgroups in Bilateral Meniere Disease. Front Neurol. 2016 Oct 24;7:182. doi: 10.3389/fneur.2016.00182. PMID: 27822199; PMCID: PMC5075646.

Author Response

Comment: This is a cross sectional survey performed with the Italian association of Meniere disease patients including 520 individuals. It is the first study on epidemiology self-reported data, and it should be published.

            Answer: We thank the Reviewer for the positive and encouraging rating of our study.

Comment:

Results

I will recommend the authors to compare their results with the studies published by Frejo et al. in 2016 and 2017 in Spain that included both around 1500 patients to obtain the distribution of the main comorbidities such as migraine or autoimmune disorders. There are 5 major clinical subgroups in Meniere disease, and it should be interesting to obtain the relative frequency of each of them.

The authors should considering to add a chart bar with the age distribution of the patients and if possible the age of onset of MD (revise Frejo el al. and compare the data).

            Answer: We thank the Reviewer for this suggestion. We have now added a whole new paragraph where we discuss in detail the results of the manuscripts by Frejo et al. (2016 and 2017) and compare them with our study results (Lines 189-206). Moreover, we have added a new set of supplementary files (Supplementary Tables 1-3) where we added the distribution of the disease onset as required by Reviewer#1 and then the age distribution and the age at onset. Finally, we have added in the Limitations section that, “..due to the way we collected and analyzed data, further comparisons with previous interesting studies on the phenotypic characteristics MD [11,25], and mainly cluster analyses, were not possible.” (Lines: 232-235)

Minor questions

Comments: The authors report familial history of 13% and allergic diseases in 33%. Could the authors compare the differences north/south within Italy since familial distribution could be different in north and south and also allergies since the respiratory allergies are related to humidity and temperatures on dry and wet seasons (Milan vs Rome).

            Answers: We thank the Reviewer for this insightful comment. Unfortunately, due to the way the data were collected and analyzed, and also due to the necessity of respecting the privacy rules and the anonymity of participants, the geographical region of participants is not a piece of information that we have available. Accordingly, we have added this aspect to the Limitations section of the discussion:  Lines 235-236.

Comment: The font size should be improved in Figure 3D and 4B, it is too tiny.

            Answer: We have now improved the font size of figures, as required.

Suggested references

Frejo L, Martin-Sanz E, Teggi R, Trinidad G, Soto-Varela A, Santos-Perez S, Manrique R, Perez N, Aran I, Almeida-Branco MS, Batuecas-Caletrio A, Fraile J, Espinosa-Sanchez JM, Perez-Guillen V, Perez-Garrigues H, Oliva-Dominguez M, Aleman O, Benitez J, Perez P, Lopez-Escamez JA; Meniere's disease Consortium (MeDiC). Extended phenotype and clinical subgroups in unilateral Meniere disease: A cross-sectional study with cluster analysis. Clin Otolaryngol. 2017 Dec;42(6):1172-1180. doi: 10.1111/coa.12844. Epub 2017 Feb 26. PMID: 28166395.

Frejo L, Soto-Varela A, Santos-Perez S, Aran I, Batuecas-Caletrio A, Perez-Guillen V, Perez-Garrigues H, Fraile J, Martin-Sanz E, Tapia MC, Trinidad G, García-Arumi AM, González-Aguado R, Espinosa-Sanchez JM, Marques P, Perez P, Benitez J, Lopez-Escamez JA. Clinical Subgroups in Bilateral Meniere Disease. Front Neurol. 2016 Oct 24;7:182. doi: 10.3389/fneur.2016.00182. PMID: 27822199; PMCID: PMC5075646.

Round 2

Reviewer 1 Report

The authors replied to all my queries.

I've no other questions

Reviewer 2 Report

The authors have improved the manuscript according to my suggestions and I have no more questions.